# A Compendium of AR Splice Variants in Metastatic Castration-Resistant Prostate Cancer

**DOI:** 10.3390/ijms24066009

**Published:** 2023-03-22

**Authors:** Khrystany T. Isebia, Martijn P. Lolkema, Guido Jenster, Ronald de Wit, John W. M. Martens, Job van Riet

**Affiliations:** 1Department of Medical Oncology and Cancer Genomics Netherlands, Erasmus MC Cancer Institute, University Medical Center Rotterdam, 3015 Rotterdam, The Netherlands; 2Division of AI in Oncology, German Cancer Research Centre DKFZ, 69120 Heidelberg, Germany

**Keywords:** androgen receptor 2, AR-Vs 3, WTS 4, prostate cancer

## Abstract

Treatment-induced AR alterations, including AR alternative splice variants (AR-Vs), have been extensively linked to harboring roles in primary and acquired resistance to conventional and next-generation hormonal therapies in prostate cancer and therefore have gained momentum. Our aim was to uniformly determine recurrent AR-Vs in metastatic castration-resistant prostate cancer (mCRPC) using whole transcriptome sequencing in order to assess which AR-Vs might hold potential diagnostic or prognostic relevance in future research. This study reports that in addition to the promising AR-V7 as a biomarker, AR45 and AR-V3 were also seen as recurrent AR-Vs and that the presence of any AR-V could be associated with higher AR expression. With future research, these AR-Vs may therefore harbor similar or complementary roles to AR-V7 as predictive and prognostic biomarkers in mCRPC or as proxies for abundant AR expression.

## 1. Introduction

Therapeutic interventions for advanced prostate cancer (PCa) are based on the inhibition of the androgen receptor (AR) and downstream of the signaling pathways. Resistance to androgen deprivation therapy is inevitable, and molecular mechanisms driving castration-resistant prostate cancer (CRPC) primarily involve alterations in AR expression and activity.

AR is a nuclear receptor transcription factor that is directly activated by androgen hormones. AR is not only considered to be a ligand-activated transcription factor, but is also able to rapidly engage and activate many signaling effectors in the cytoplasm of target cells, leading to cell cycle progression, invasion, cytoskeleton changes, and so on [1,2]. The gene encoding AR is located on the X chromosome on locus Xq11-Xq12. The protein structure of AR is similar to other members of the nuclear receptor family, consisting of the N-terminal domain (NTD), followed by the DNA-binding domain (DBD), and the C-terminal ligand-binding domain (LBD), which is connected to the DBD by a flexible hinge region. After testosterone becomes converted by 5-α-reductase to the potent dihydrotestosterone (DHT), the ligand-binding pocket binds the DHT, and the AR translocates into the nucleus, in which it binds to androgen response elements (AREs) [3].

The LBD plays a key role in the molecular mechanism of AR activity, and persistent AR activity has been extensively explored as a mechanism linked to AR splice variants (AR-Vs). Several of these AR-Vs lack the LBD, thus promoting constitutive gene transcription in the absence of androgen ligands [4]. To this date, many AR-Vs have been reported in CRPC specimens. Importantly, since the androgen signaling remains active even in the CRPC stage, this has led to the clinical development of androgen receptor signaling inhibitors (ARSI), targeting the LBD of AR directly and indirectly. Several studies [5,6,7,8,9,10] have shown that the most frequently observed and clinically relevant AR-V within CRPC settings is AR-V7. AR-V7 has been associated with primary resistance to abiraterone or enzalutamide therapy in metastatic CRPC and, for this reason, has been extensively explored [6]. To date, the majority of articles available mostly focus on AR-V7 and AR-V9, utilize cfRNA, or include a small number of CRPC samples [6,8,9]. Whilst these studies have advanced our understanding of these enigmatic AR-Vs in relation to disease context, a larger cohort of late-stage mCRPC patients might elucidate the most relevant AR-Vs for subsequent experimental characterization.

Therefore, our aim was to uniformly determine recurrent AR-Vs in a large cohort of mCRPC patients, using whole transcriptome sequencing in order to re-evaluate the landscape of recurrent AR-Vs in mCRPC patients in order to prioritize subsequent research. Our unique cohort of WTS analyses of tissue biopsies allows us to provide a complete landscape of AR splicing.

## 2. Results

### 2.1. Compilation of a Uniform Compendium of AR Splice Variants (AR-Vs) in mCRPC

We utilized the whole-transcriptome sequencing (WTS) data derived from needle biopsies of 278 distinct patients with metastatic castration-resistant prostate cancer (mCRPC) from the CPCT-02 study to inventory the expression of recurrent AR splice variants (AR-Vs; Figure 1a).

Baseline characteristics of the included patients from the CPCT-02 study (*n* = 278) with available clinical records (*n* = 213) are reported in Table 1. The median age at biopsy of the included patients was 67 years, with a range between 46 and 85 years. This multicenter cohort consisted of mCRPC patients undergoing a wide range of treatments. On median, patients received a total of two treatment lines prior to biopsy. The majority (82.6%) of patients were pre-treated with docetaxel, and more than half (52.6%) were pre-treated with enzalutamide. In addition, 73 patients were included who were only treated with a single treatment (next to ADT) prior to biopsy comprising of docetaxel (*n* = 48), enzalutamide (*n* = 18), Radium-233 *(n* = 3), abiraterone (*n* = 2), carboplatin (*n* = 1), and estramustine (*n* = 1).

In addition, we utilized a publicly-available WTS dataset comprising 89 biopsies from non-matched hormone-sensitive prostate cancer (hsPC) patients prior to ADT, comprising both prostate tumor tissue (*n* = 49) and normal-adjacent prostate tissue constituting mostly normal-adjacent tissue (NAP; *n* = 40).

Within the CPCT-02 WTS samples, we inventoried all recurrently-expressed splice junctions overlapping the AR locus to generate a uniform compendium of canonical exons, potential cryptic exons, and 5′ UTR shortenings as observed within late-stage mCRPC. This compendium was supplemented with additional (known) cryptic exons from the literature [3] to also investigate additional AR splice variants (AR-Vs) found previously in other studies (Appendix A).

### 2.2. Quantification of the Relative Expression of AR-Vs in mCRPC

Using the supporting spliced reads for each splice junction distinctive to each AR-V and comparing this against the support for the canonical splice junction between the first two canonical exons of AR, we calculated the sample-wise relative expression of each AR-V within our cohort (Figure 1b). From least to most expressed (based on median relative expression per AR-V) within our comprehensive mCRPC cohort, we could observe evidence of AR45 (median, Q1-Q3; 1.56, 0–1.65), AR-V3 (0.82, 0–0.95), AR-V7 (0.55, 0–1.17), AR23 (0.2, 0–0.43), AR-V1 (0, 0–0.25), and AR-V9 (0, 0–0.34), whilst all other previously described AR-Vs (*n* = 10; Appendix A) lacked expression. AR-V11 was exempt from this analysis as no distinctive splice junction could be used for quantification.

When evaluating for minimal expression (relative expression of ≥1% compared to AR), we noted that certain AR-Vs could be more prevalently observed than others: AR45 (*n* = 184; 66%), AR-V3 (*n* = 113; 41%), AR-V7 (*n* = 103; 37%), AR-V9 (*n* = 33; 12%), AR23 (*n* = 23; 8%), and AR-V1 (*n* = 18; 6%). AR45 was observed in the majority of the samples.

In addition, we did not observe mutual exclusivity of expressed AR-Vs within our mCRPC cohort (Figure 1c).

### 2.3. Quantification of the Relative Expression of AR-Vs in Treatment-Naïve Primary Prostate Cancer and Normal-Adjacent Tissue

When performing the identical analysis using the same compendium of AR-Vs and canonical exons on the treatment-naïve PCa and NAP samples from our NGS-ProToCol cohort, we only observed supporting splice junctions for all canonical AR exons (exons 1–8) and no support for distinct splice junctions for any AR-V, incl. AR-Vs which were also not observed in our mCRPC cohort. Therefore, the absence of AR-Vs in treatment-naïve (hormone-sensitive PCa) and NAP samples points toward the emergence of AR-Vs due to treatment or disease progression rather than occurring within early-stage malignant or normal-adjacent tissue.

### 2.4. Differences in Observed AR-Vs Compared to Total AR Expression

We evaluated whether harboring expression (≥1%) of any AR-Vs was associated with more abundant expression of AR, in general. We observed statistically significant differences in which samples that harbored an AR-V indeed displayed more abundant expression of AR (Figure 2a; two-sided Mann–Whitney test; *p* < 0.001).

We also evaluated whether the level of AR expression could be associated with the number of distinct expressed AR-Vs, but this lacked statistical significance (Figure 2b).

### 2.5. Differences in Observed AR-Vs in Comparison to Clinical Characteristics

To perform exploratory analysis into possible trends with relative AR-V expression (tested per AR-V) versus clinical characteristics, we first performed univariate regression of age at biopsy, given treatment line at any moment prior to biopsy (docetaxel, enzalutamide, cabazitaxel, or Radium-223), and the total number of treatment lines to determine potentially significant univariate features (*p* < 0.05). From these univariate analyses, the total number of treatment lines (AR45 and AR-V9) and treatment with enzalutamide (AR-V1 and AR-V7), cabazitaxel (AR45), and Radium-223 (AR23 and AR-V3) had significant interactions with the relative expression of specific AR-Vs.

Next, we performed multivariate regression analysis using these four features within the same model and retained significant trends (*p* < 0.05) with AR-V specific relative expression between cabazitaxel treatment and AR45 (*p* = 0.0265), enzalutamide treatment and AR-V7 (*p* = 0.0276), and Radium-233 with AR-V3 (*p* = 0.0499). These effects were therefore seen as significant whilst correcting for the total number of treatment lines within the model.

As these prior treatments constituted a heterogenous mixture of treatments, we performed a similar analysis within the substantial subset of patients who received only a single line of treatment of either docetaxel (*n* = 48) or enzalutamide (*n* = 18) next to ADT. Between these two patient groups, we observed univariate significance (*p* < 0.05) of age at biopsy with relative expression of AR-V3 and AR-V7, enzalutamide treatment with AR-V7 and AR-V9, and docetaxel with AR-V7. Next, we performed multivariate regression analysis utilizing all three features and observed the retainment of enzalutamide treatment with both the relative expression of AR-V9 (*p* = 0.0104) and AR-V7 (*p* = 0.0157), as shown in Figure 3. In addition, a trend was also observed between docetaxel pre-treatment and AR23 (*p* = 0.0381; Figure 3).

## 3. Discussion

Numerous studies [5,6,7,8,9,10] have been undertaken to study molecular mechanisms driving castration-resistant prostate cancer. From this, persistent AR activity has been extensively explored as a mechanism linked to AR splice variants (AR-Vs). AR-Vs could play a potential role in primary and acquired resistance to conventional and next-generation hormonal therapies in prostate cancer and therefore have gained momentum [11].

We utilized the WTS data of 278 distinct patients with mCRPC, together with 49 hormone-sensitive prostate cancer tissues and 40 normal-adjacent prostate tissues, to inventory the expression of recurrent AR-Vs and generate a uniform compendium of AR-Vs in both hormone-sensitive and late-stage castration-resistant setting. A total absence of any AR-V was observed within the hsPC tissues, whilst the mCRPC tissues revealed expression of various AR-Vs. This may possibly point toward the emergence of AR-Vs due to treatment or disease progression [5,9,12].

From this compendium, we found that AR45 was the most frequently expressed AR-V in our mCRPC cohort (based on ≥1% relative expression compared to AR), followed by AR-V3 and AR-V7. However, we also observed expression of AR-Vs below this threshold, meaning that our currently selected threshold may be under-representing for the actual occurrence of an AR-V, yet this allowed us to better capture highly-expressed AR-Vs from lowly-expressed signals. AR45, as named after its molecular mass of 45 kDa, has previously been described as an enigmatic factor in various mammalian tissue, which uniquely retains the LBD and either represses or co-activates AR activity depending on certain co-factors and active pathways [13,14,15,16,17]. Its specific role in mCRPC is therefore yet to be determined, but as the most abundantly-expressed AR-V, this warrants additional investigation.

With our current cohort, we cannot robustly discern the potential roles of AR-Vs within this disease context due to the large heterogeneity of treatment lines and included patients. In addition, we also lacked detailed clinical characteristics such as the Gleason score and histological features. Therefore, characterizing the exact role of AR-Vs with dual roles such as AR45, known as both a repressor or pro-oncogenic factor of AR activity [13], would necessitate experimental follow-up studies. As our results are based on bulk sequencing, we cannot discern the exact cellular origin of the AR-Vs (e.g., malignant cells or stroma). However, we can detail the occurrence and relative expression of AR-45 and other AR-Vs and propose several of these to be recurrent within mCRPC. This can help prioritize follow-up research into the top recurrent AR-Vs in mCRPC.

Of these recurrent AR-Vs, AR-V7 has shown much potential as a predictive and prognostic biomarker in mCRPC [18,19]. From our findings, perhaps not only AR-V7 might serve such a function, but also AR-45, AR-V3, and AR-V9, as they are recurrently expressed in mCRPC and seem to increase with expression per number of treatment lines given. As these AR-Vs were absent within our treatment-naive patients, these AR-Vs could likewise play a direct or indirect role in castration-resistance in mCRPC due to given treatment and could serve as relevant biomarkers for treatment resistance of disease progression. We partly observed this within patients with only a single treatment line and ADT prior to biopsy, as we observed higher relative expressions of AR-V7 and AR-V9 in enzalutamide-treated mCRPC patients compared to docetaxel-treated mCRPC patients and, conversely, AR23 expression in docetaxel-treated patients.

We also report that the majority of mCRPC patients expressed one or more AR-Vs in a non-mutually exclusive (co-expressed) manner [6]. Furthermore, we found that the presence of AR-Vs was related to AR expression and could therefore be an important proxy for abundant AR expression resulting from various sources of somatic aberrations such as genomic amplification of AR. Such a proxy might be interesting as AR-amplified cancers have been shown to respond better to second-line maximal androgen blockade compared to tumors without AR amplification.

In conclusion, this uniform inventory of AR-Vs in mCRPC shows that next to AR-V7, AR45, AR-V3, and AR-V9 are also recurrently observed and that future research might further establish their biological roles within mCRPC. Additional studies could furthermore establish the potential of these AR-Vs as predictive and prognostic biomarkers in mCRPC or as a proxy for abundant AR expression due to various somatic aberrations.

## 4. Materials and Methods

### 4.1. Patient Inclusion and Clinical Parameters

As part of the CPCT-02 Dutch nationwide multicenter trial (NCT01855477), patients with local prostate or metastatic prostate malignancies were eligible for inclusion and subsequent sampling for whole-transcriptome sequencing if the following criteria were met: (1) age ≥ 18 years; (2) indication for a new line of systemic treatment with registered anti-cancer agents; and (3) safe biopsy according to the intervening physician. All patients have given explicit informed consent for sequencing, data sharing for cancer research purposes, the collection of matched peripheral blood samples (reference DNA), and image-guided percutaneous biopsy from a malignant lesion. For the current study, patients were included who underwent biopsy between 18 February 2015 and 6 June 2020, resulting in a cohort of 278 distinct mCRPC samples. As a reference dataset, we interrogated the whole-transcriptome sequencing data from patients with primary prostate cancer prior to androgen deprivation therapy (ADT) from the NGS-ProToCol cohort (EGAD00001004215) [20], comprising both prostate tumor tissue (*n* = 49) and normal-adjacent prostate tissue (NAP; *n* = 40). The primary prostate cancer patients included prior to ADT are considered hormone-sensitive prostate cancer patients, whilst the normal-adjacent prostate (NAP) samples can be considered prostate cancer stroma. We used both groups of patients as our reference dataset to compare AR-Vs in distinct prostate cancer states.

### 4.2. Sample Acquisition and Whole-Transcriptome Sequencing

Whole-transcriptome sequencing of the CPCT-02 cohort (*n* = 278 biopsies from 278 distinct patients; one from each patient) was performed according to the manufacturer protocols using a minimum of 100 ng total RNA input. Total RNA was extracted using the QIAsymphony RNA kit (QIAGEN, FRITSCH GmbH, Idar-Oberstein, Germany) according to manufacturer instructions. Paired-end sequencing of (m)RNA was performed on either the Illumina NextSeq 550 platform (2 × 75 bp; Illumina, San Diego, CA, USA) and NovaSeq 6000 platform (2 × 150 bp; Illumina, San Diego, CA, USA), depending on the time of sequencing.

### 4.3. Pre-Processing and Alignment of Raw Reads

Prior to alignment, raw reads (per lane) from the CPCT-02 samples were pre-processed using fastp [21] (v0.23.2) to trim adapter sequences (paired-end), low-quality bases and perform low-complexity trimming but without a min. length selection on the remainder of the read. Subsequently, these corrected reads are aligned against the human reference genome (GRCh37) with GENCODE [22] (v38) annotations using STAR (v2.7.9a) [23]. Per sample, all lanes (both R1 and R2; paired-end) were used during alignment using the following command:

STAR --genomeDir <GRCh37> --readFilesIn <R1 lanes> <R2 lanes> --readFilesCommand zcat --outFileNamePrefix <prefix> --outSAMtype BAM Unsorted --outSAMunmapped Within --outSAMattributes NH HI AS nM NM MD jM jI MC ch XS --outSAMstrandField intronMotif --outFilterMultimapNmax 10 --outFilterMismatchNmax 3 --limitOutSJcollapsed 3,000,000 --chimSegmentMin 10 --chimOutType WithinBAM SoftClip --chimJunctionOverhangMin 10 --chimSegmentReadGapMax 3 --chimScoreMin 1 --chimScoreDropMax 30 --chimScoreJunctionNonGTAG 0 --chimScoreSeparation 1 --outFilterScoreMinOverLread 0.33 --outFilterMatchNminOverLread 0.33 --outFilterMatchNmin 35 --alignSplicedMateMapLminOverLmate 0.33 --alignSplicedMateMapLmin 35 --alignSJstitchMismatchNmax 5 -1 5 5 --twopassMode Basic --twopass1readsN -1 --runThreadN 15 --quantMode TranscriptomeSAM --outSAMattrRGline <sample-specific readgroup>

Post-alignment, duplicate reads were marked using sambamba markdup [24] (v0.8.1), and general alignment metrics (e.g., number of primary-mapped reads) were retrieved using sambamba flagstats (v0.8.1). Read counting of primary alignment was performed using featureCounts (v2.0.3) using GENCODE (v38) annotations [25].

The whole-transcriptome sequencing data of NGS-ProToCol was processed as previously described [26]. In brief, sequence adapters (TruSeq3) were trimmed using Trimmomatic(v0.38), and paired-end reads were subsequently aligned to the human reference (GRCh38.p12) using STAR [23] (v2.7.0a) with genomic annotations from GENCODE [22] release 29. Generation of alignment quality metrics (flagstat) and duplicate reads marking was performed by sambamba [24] (v0.6.7).

### 4.4. Detection and Reconstruction of Splice Junctions within AR

All primary-aligned, properly paired, and non-duplicated spliced alignments overlapping with a predefined genomic locus representing the AR locus (GRCh37; chrX:66763863-66951462;+) were retrieved (with strandness) from the Binary Alignment Map (BAM) files using standardized BioConductor approaches in R [27]. These spliced alignments were used to determine recurring splice junctions (SJ), which represent the genomic locations of recurring exonic boundaries, i.e., the canonical and potential cryptic exons within AR. Splice junctions with ≥5 (strand-correct) spliced reads, which were observed in ≥10 samples, were used to designate recurring observable exonic borders. These locations were supplemented with known information on cryptic exons from the literature [28] to establish a compendium of canonical, cryptic exons and 3′ UTR shortenings with genomic locations (GRCh37), as shown in Appendix A. These genomic locations were lifted-over to GRCh38 to acquire the splice junction coverages for the NGS-ProToCol cohort.

### 4.5. Quantification of the Relative Expression of AR-Vs

To determine the sample-wise and relative expression of any given AR-V, we used the number of splice junction reads supporting the canonical exons 1 and 2 of AR as a reference to determine the following ratio:

Equation (1):(1)$$y=\frac{SJ_{E1E2}}{SJ_{x}}$$

Here, *SJ_E_*_1*E*2_ is the number of primary-aligned, properly paired, non-duplicated spliced alignments (reads) supporting the splice junction between Exon 1 and Exon 2 and *SJ_x_* being the similar splice junction support for the AR-V-specific splice junction *x*, as shown within Appendix A.

### 4.6. Association of Relative Expression of AR-Vs to Clinical Characteristics

To determine possible correlations or interactions between the relative expression of AR-Vs with clinical characteristics such as age at biopsy, the total number of treatment lines, and specific treatments, we performed univariate and multivariate regressions on all complete patient cases (patients without missing clinical data; *n* = 213).

We first performed univariate regression of age at biopsy, the total number of treatment lines, and major treatment lines (treatments decoded as 1 for received or 0 if not received) on all evaluable patients (*n* = 213) to determine the univariate significance of these metrics in regards to AR-V expression, in which each AR-V was tested separately. For example, we used a univariate regression model with age at biopsy versus relative expression of AR45. Variables with *p* < 0.05 were considered to be significant contributors to relative AR-V expression.

Next, we subsetted our cohort for patient groups with only a single line of treatment prior to biopsy. In order to robustly determine potential interactions, we only retained patient groups with a min. of 10 patients per group, which contained patients treated with docetaxel (*n* = 48) and enzalutamide (*n* = 18) prior to biopsy.

Using these patients (*n* = 66), we performed a similar univariate analysis in which age at biopsy, enzalutamide treatment, and docetaxel treatment (treatments decoded as 1 for received or 0 if not received) were compared to relative AR-V expression, again per AR-V. Variables with *p* < 0.05 were considered to be significant contributors to relative AR-V expression. In addition, a multivariate analysis per AR-V with all statistically significant features from the univariate analysis was combined into a multivariate analysis (age at biopsy, enzalutamide, and docetaxel treatment), and similarly, a threshold of *p* < 0.05 was used to determine statistical significance to relative AR-V expression.

### 4.7. Data Availability

The WTS and corresponding clinical data used in this study were made available by the Hartwig Medical Foundation (HMF; Dutch nonprofit biobank organization) after signing a license agreement stating data cannot be made publicly available via third-party organizations. Therefore, the data are available under restricted access and can be requested by contacting the Hartwig Medical Foundation (https://www.hartwigmedicalfoundation.nl/en/data/data-acces-request/ (accessed on 30 January 2023) under the accession code DR-071. The raw sequencing data of the NGS-ProToCol was previously deposited on the European Genome-phenome Archive (EGA) under accession number EGAS00001002816. The remaining data are available within this article, Appendix A, or available from the authors upon request.

### 4.8. Code Availability

All custom code and scripts used within this study (processing, analysis, and visualization) have been deposited on Zenodo/GitHub under the GPL-3.0: https://zenodo.org/record/7585871

## Figures and Tables

**Figure 1 ijms-24-06009-f001:**
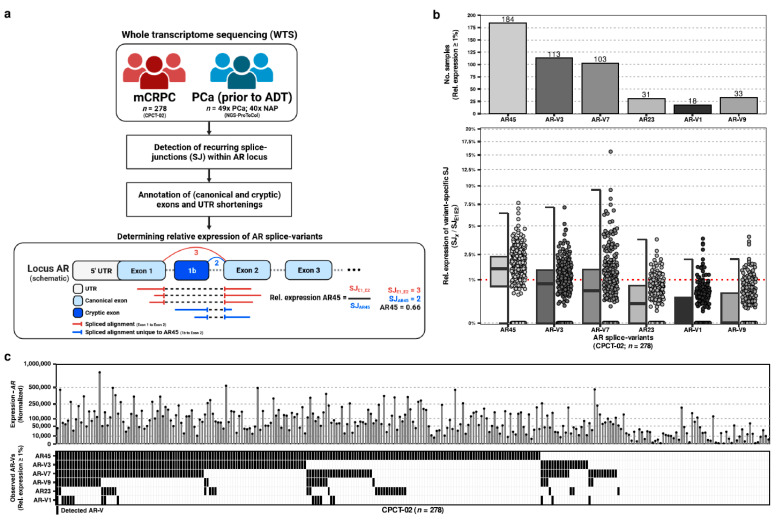
Overview of this study and the expression of AR-Vs in the CPCT-02 mCRPC cohort. (**a**) Schematic overview of this study and the methodology to determine expression of AR-Vs; (**b**) The top panel depicts the number of samples from the CPCT-02 (*n* = 278) with relative expression of AR-V specific SJ ≥1% (compared to canonical AR exon 1 to exon 2). The lower panel depicts the relative expression per AR-V for all CPCT-02 samples. Boxplots represent the median, first, and third quantile whilst error bars depict the interquartile range (IQR) x.5; (**c**) Overview of expressed AR-Vs (≥1% relative expression) per sample ordered on mutual exclusivity. The top panel depicts the normalized expression of AR (all exons).

**Figure 2 ijms-24-06009-f002:**
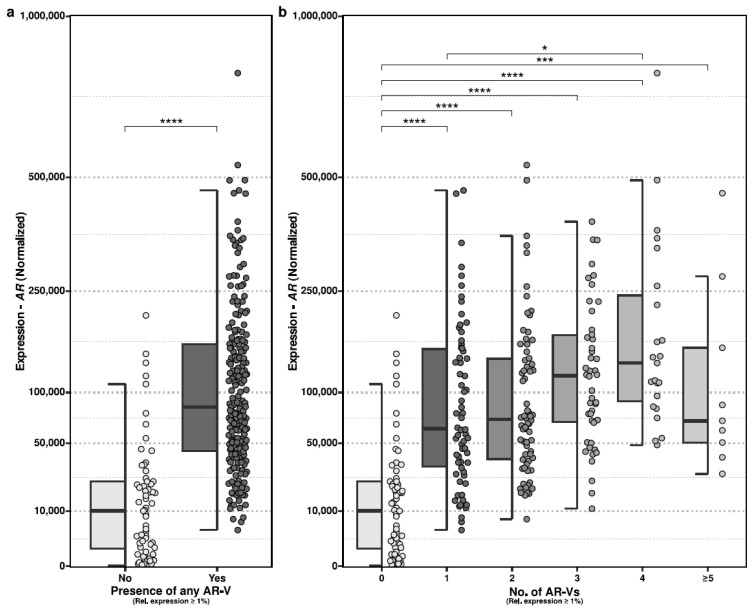
Presence of AR-Vs compared to expression of AR. 9. Normalized expression of AR (*y*-axis; square-root transformed) between mCRPC samples (CPCT-02) with and without expression of any AR-V (≥1% relative expression). Statistical significance (*p* value) is denoted between groups (Mann –Whitney test) *p* values is denoted as followed: **** *p* < 0.0001, *** *p* < 0.001, * *p <* 0.05 (similar for adjusted *p*). (**b**) Similar to (**a**), but categorized per number of expressed AR-Vs. Statistical significance (*q* value) is denoted between groups (two-sided Mann–Whitney test with Benjamini–Hochberg correction).

**Figure 3 ijms-24-06009-f003:**
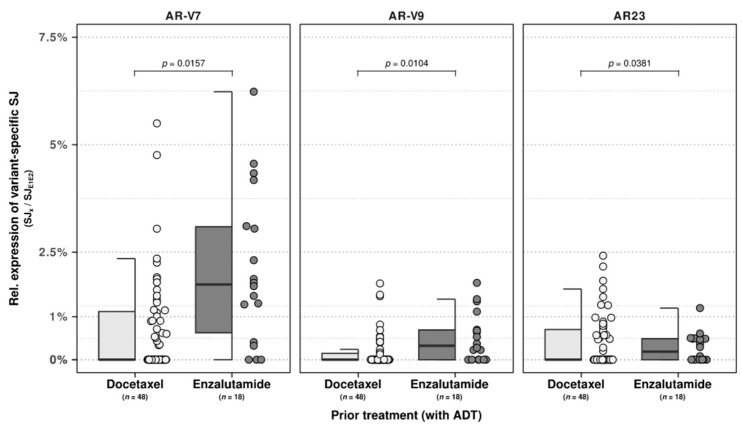
The occurrence and relative expression of AR-Vs can be attributed to distinct pre-treatment lines. Relative expression of AR-Vs versus AR (*y*-axis) between mCRPC samples (CPCT-02) pre-treated with a single treatment line of docetaxel or enzalutamide, together with androgen deprivation therapy (ADT). Statistical significance of the pre-treatment line from the multivariate regression model (*p* value), including age at biopsy and whether patients were treated with either docetaxel or enzalutamide, is denoted between groups.

**Table 1 ijms-24-06009-t001:** Baseline characteristics of included mCRPC patients (CPCT-02).

Characteristic	Total (*n* = 278)	Missing (no.)
**Age at registration**, Median (range)–yr	67 (46–85)	2
**Total prior treatment lines**, Median (range)–*n*	2 (1–8)	65
**Type of prior treatment**, no. (%)		
-Abiraterone	50 (23.5%)	65
-Enzalutamide	112 (52.6%)	65
-Docetaxel	176 (82.6%)	65
-Cabazitaxel	55 (25.8%)	65
-Radium-223	28 (13.1%)	65
-Other second ARSI	2 (0.9%)	65
-Other chemotherapy	8 (3.8%)	65
-Other treatment	4 (1.9%)	65
*ARSI = Androgen Receptor Signaling Inhibitors*		

## Data Availability

Restrictions apply to the availability of these data. Data was obtained from HMF and are available at https://www.hartwigmedicalfoundation.nl/en/data/data-acces-request/ (accessed on 30 January 2023) with the permission of HMF.

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
