# Peer review of "A Compendium of AR Splice Variants in Metastatic Castration-Resistant Prostate Cancer"

_ijms, 2023, doi:10.3390/ijms24066009_

Round 1

Reviewer 1 Report

1. Please show why your article is novel. Similar papers have been published by others (Fettke H, et al. Eur Urol. 2020 Aug;78(2):173-180 and others). 

2. AR-45 is the opposite of AR-V7 and other variants - in that it lacks the N-terminal domain and not the ligand binding domain. AR-45 inhibits the function of full-length AR and slows down cell growth (Ahrens-Fath I, et al. FEBS J. 2005 Jan;272(1):74-84). How does this information work with the concept that it is a marker of aggressive prostate cancer? 

3. AR-45 data needs to be compared to hormone sensitive prostate cancer and non-tumor prostate data. 

Reviewer 2 Report

No comments on this manuscript.

Reviewer 3 Report

Castrate resistance remains a disease of unmet clinical need and AR variants appear to be important in therapy failure.  The identification of AR variants and their role in disease progression is therefore of significant interest.  In this study, the authors investigate AR variant expression in normal, treatment naive and therapy resistant prostate cancer.  The study is sound and interesting data are presented, however the volume of data and the novelty is limited.  For example, a number of other studies have also investigated AR variant expression in patient samples and some of these are lacking from the introduction e.g. DOI: 10.1038/s41591-021-01244-6

I therefore feel that the intro/discussion should be modified to provide a more comprehensive review of previous studies.  I also feel that some additional analyses be performed to increase the novelty of the research.  For example, it would be interesting to see if the expression of specific coactivators/co-repressors/signalling (as mentioned in the discussion) correlates with AR54 expression e.g. perhaps there will be a switch to pro-oncogenic AR54 signalling in CRPC?  

Reviewer 4 Report

The manuscript by Dr Khrystany T. Isebia and colleagues is focused on the analysis of AR and its splicing variants in a population of patients affected by CRPC.  Authors consider the whole-transcriptome sequencing data derived from biopsies of 278 distinct patients with metastatic castration-resistant prostate cancer (mCRPC), 89 biopsies from (non-matched) prostate cancer patients prior to ADT; comprising both prostate tumour tissue (n = 49) and normal-adjacent prostate tissue (NAP; n = 40).

I have some questions:

1. The data about the expression of AR45 in CRPC or Patients before ADT or unaffected population is not immediate. Is the correlation between the therapy and the onset of this splicing variant?

2. Have the authors also data about the AR variants expressed in PC stroma? They should at least discuss these findings.

3. is there a correlation between these splice variants and the therapies, the age, the Gleason score, and the acquisition of a neuroendocrine phenotype? The authors should add details or introduce their data better.

4. A table resuming more characteristics of the patients could be appreciated. 

Reviewer 5 Report

In this manuscript the authors analyze the frequency of expression of androgen receptor variants in 278 mCRPC biopsies.

The topic of this work is very interesting and highlights an important aspect of the chemoresistance in prostate cancer: the role of AR splicing variants in these advanced forms of prostate cancer.

Considering this, the manuscript should be made clearer, mostly in Introduction and in Discussion and should be emphasized the aim that the authors want reach. The authors should explain where and how they used the data about the normal prostate samples and the PC before the therapy. They write” In addition, we utilized an additional publicly-available WTS dataset comprising 89 biopsies from (non-matched) prostate cancer patients prior to ADT; comprising both prostate tumor tissue (n = 49) and normal-adjacent prostate tissue (NAP; n = 40).” In the manuscript is never clear when these data are used.

Do the authors compare the AR-Vs expression in normal tissues and in PC samples before androgen deprivation therapy?

It would be interesting correlates the AR Vs expression and the PC tumor grading or, at least, discuss about it.

Round 2

Reviewer 1 Report

The authors have addressed all major concerns. 

Author Response

The authors would like to thank the reviewer for his/her valuable comments. 

Reviewer 4 Report

Whether there is not a correlation between the reviewers and the files uploaded (as rebuttal letter), the manuscript is improved in this form.

Author Response

(The authors gave the same response as above.)

Reviewer 5 Report

The authors reply to all my comments. In this form, the manuscript can be published on this journal.

Author Response

(The authors gave the same response as above.)
